# Sketch, capture and layout phylogenies

**Daniel H. Huson** *

Institute for Bioinformatics and Medical Informatics, University of Tübingen, Tübingen, Germany

* daniel.huson@uni-tuebingen.de

## Abstract

Phylogenetic trees and networks play a central role in biology, bioinformatics, and mathematical biology, and producing clear, informative visualizations of them is an important task. We present new algorithms for visualizing rooted phylogenetic networks in either a "combining" or "transfer" view, in both cladogram and phylogram style. In addition, we introduce a layout algorithm that aims to improve clarity by minimizing the total reticulate displacement of reticulate edges. To address the common issue that biological publications often omit machine-readable representations of depicted trees and networks, we also provide an image-based algorithm that assists in extracting their topology from figures. All algorithms are implemented in our new open source PhyloSketch app.

## Author summary

Phylogenetic trees and networks provide visual representations of evolutionary relationships, but creating clear and accurate diagrams can be challenging, especially when evolutionary histories involve hybridization, recombination, or horizontal gene transfer. We present PhyloSketch, an interactive app for sketching, capturing, and improving the layout of phylogenetic trees and networks. The program introduces visualization algorithms that display evolutionary relationships in alternative "combining" and "transfer" views, and in both cladogram and phylogram styles. A novel layout algorithm enhances clarity by minimizing the displacement of reticulate edges—connections that represent non-tree-like evolution. To help make published results more reproducible, PhyloSketch also includes an image-based tool that assists users in capturing network structures from figures. Together, these features make it easier to produce, analyze, and share high-quality visualizations of evolutionary relationships in research and education.

## 1 Introduction

Rooted phylogenetic networks are used to represent putative evolutionary scenarios in the presence of reticulate events such as horizontal gene transfer, hybrid speciation, or reassortment.

**Data availability statement:** Code is open source, available here: https://github.com/husonlab/phylosketch2.

**Funding:** This publication is based upon work supported by the National Science Foundation under Grant No. DMS-1929284 while the author was in residence at the Institute for Computational and Experimental Research in Mathematics in Providence, RI, during the "Theory, Methods, and Applications of Quantitative Phylogenomics" program. This work was also supported by the NZ Marsden Fund (23-UOC-003) during a visit to New Zealand. The funders had no role in study design, data collection and analysis, decision to publish, or preparation of the manuscript.

**Competing interests:** The authors have declared that no competing interests exists.

Several algorithms exist for computing such networks from biological data [1–9], typically producing a description in the extended Newick format [10]. While any phylogenetic tree can be embedded in the plane without edge crossings in linear time, a rooted phylogenetic network may not be planar. In such cases, the computational problem arises of finding an embedding that minimizes the number of edge crossings caused by reticulation. This problem is known to be NP-hard, even for binary networks interpreted as transfer networks [11].

Although there are many widely-used tools for interactively drawing and editing phylogenetic trees [12,13], there are few that also target phylogenetic networks [14–18].

Dedicated algorithms are required to visualize rooted phylogenetic networks. Here, we present backbone-tree-based algorithms for computing embeddings of rooted phylogenetic networks, targeting combined and transfer cladograms and phylograms (see Fig 2). We also provide a heuristic aimed at minimizing the reticulate displacement of reticulation edges – a problem that is NP-hard in general, as it solves the Minimum Linear Arrangement Problem [19]. In practice, we address it using a simulated annealing approach [20].

We introduce the PhyloSketch App, an interactive program for sketching rooted trees and networks, computing their layout using the new algorithms, and importing/exporting in extended Newick format (see Fig 1).

We also describe a simple image-based workflow implemented in PhyloSketch that helps users extract rooted phylogenetic trees and networks from figures. This addresses a common problem in bioinformatics: biology publications often fail to provide a machine-readable description of the topology of the trees or networks they depict. For example, the phylogenetic network on lizard evolution shown in Fig 2A [22] is only available as an image in the paper. (However, in this particular case, the Newick string was obtainable upon request from the first author.)

Previous approaches [23,24] to capture have been limited to phylogenetic trees.

All algorithms are provided in the PhyloSketch App, which is open source and available at: https://github.com/husonlab/phylosketch2.

## 2 Results

The main results focus on the layout of rooted phylogenetic trees and networks.

### 2.1 Rooted phylogenetic networks and their layout

Let $X$ be a set of taxa. A rooted phylogenetic network $N = (V, E, \rho, \lambda)$ on $X$ is a directed graph with node set $V$, edge set $E \subseteq V \times V$, root node $\rho \in V$, and taxon labeling $\lambda$, such that [26]:

1. The graph $N$ is a directed acyclic graph (DAG),
2. There is exactly one root node with in-degree 0, namely $\rho$ (and thus, in particular, $N$ is connected),
3. The labeling $\lambda : X \to V$ is a bijection between $X$ and the set of leaves (nodes of out-degree 0),

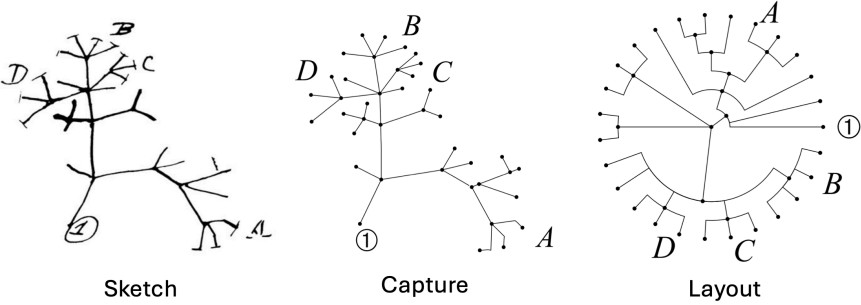

**Fig 1. Graphical abstract.** Sketch courtesy of Cambridge University Library [21].

4. There are no "through nodes" that have both in-degree 1 and out-degree 1, and
5. All leaves should have in-degree $\leq 1$.

A node $v$ is called a *tree node* if it has in-degree $\leq 1$, and a *reticulate node* otherwise. An edge $e = (v, w)$ is called a *tree edge* or *reticulate edge* if its target node $w$ is a tree node or reticulate node, respectively.

We call a network *bifurcating* if all nodes $v$ have out-degree $\deg^+(v) \leq 2$, and *bicombining* if all nodes $v$ have in-degree $\deg^-(v) \leq 2$.

If there are no reticulate edges, then $N$ is a tree, and thus planar, it can be drawn in the plane without edge crossings. In the presence of reticulate edges, $N$ may be non-planar, and any drawing of $N$ may necessarily involve crossings.

Viewing a rooted phylogenetic network as a generalization of the widely-used concept of a rooted phylogenetic tree, our goal when drawing a non-planar rooted phylogenetic network is to do so in such a way that *tree edges never cross each other*, while *reticulate edges may cross any edges*. To enhance visual clarity, we aim to reduce the number and extent of crossings in the drawing.

[11] investigates the problem of minimizing the number of crossings in bifurcating and bicombining rooted phylogenetic networks. They assume that the network $N$ is tree-based (i.e., possesses a spanning tree $T$, which is assumed to be provided as part of the input), and study theoretical properties of different ways to draw the network in a "transfer view" (as defined below).

Here, we consider the more general case in which networks are not necessarily bifurcating or bicombining, and reticulations can be displayed either in a *combining view*, a *transfer view*, or using a mixture of both. Rather than explicitly trying to minimize the number of crossings, we instead aim to minimize the total *reticulate displacement* of reticulate edges in a left-to-right drawing of the network.

## 2.2 Combine view

There are two distinct ways to interpret reticulate nodes in a rooted phylogenetic network. In a *combining view*, a reticulate node represents a *combining event*, such as *hybridization-by-speciation*, where all incoming edges are treated equally and rendered in a similar fashion (see Fig 2B). In contrast, a *transfer view* represents a *transfer event*, such as *horizontal gene transfer*. In this case, one incoming edge –the *transfer-acceptor edge*– represents the main lineage and is drawn like a regular tree edge, while all other incoming edges represent transferred material and are drawn as reticulate edges (see Fig 2D).

While the visualization of a rooted phylogenetic network can easily accommodate the simultaneous occurrence of both combining and transfer nodes, when drawing cladograms, two different strategies appear to lead to better results depending on whether the network mainly contains combining or transfer nodes.

## A. Captured hybridization network

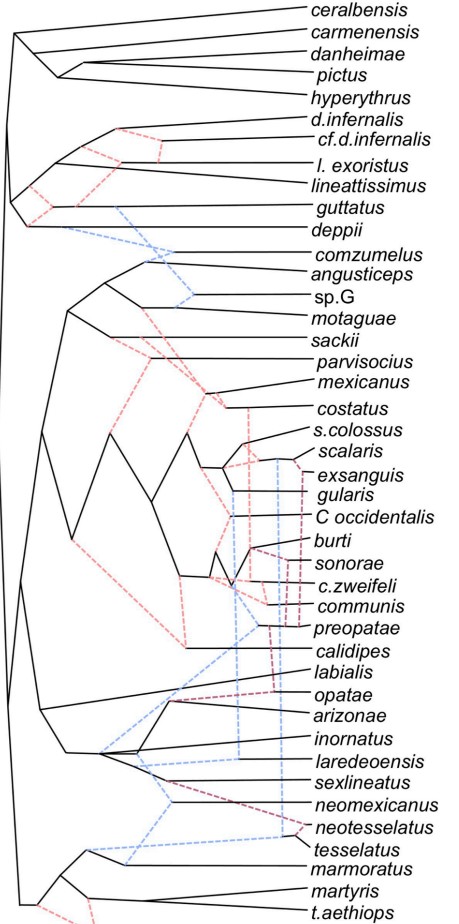

## B. Combining view of network

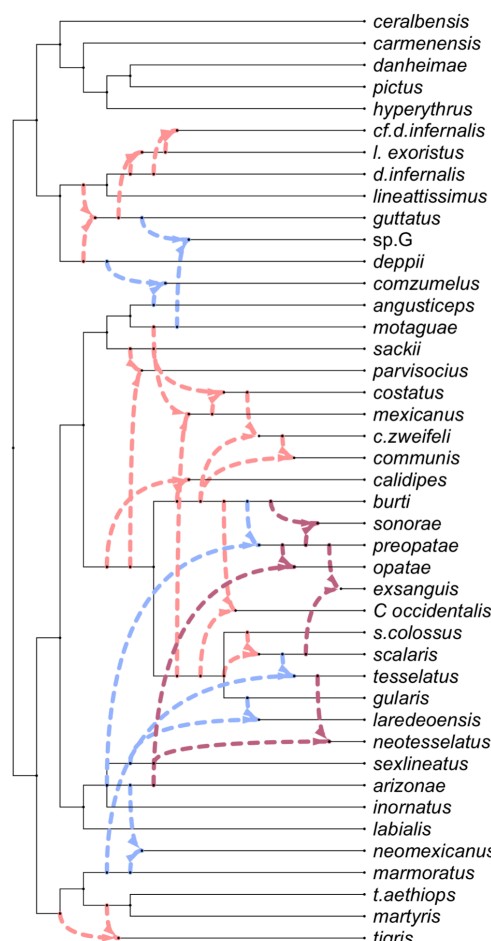

## C. Captured transfer network

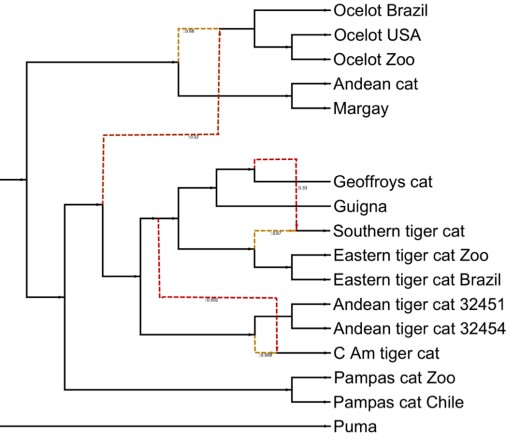

## D. Transfer view of network

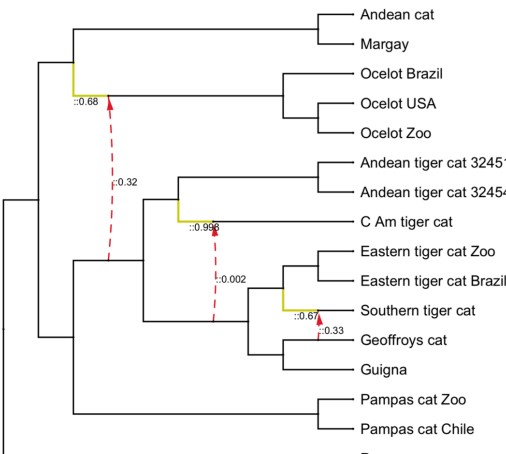

**Fig 2. Capture and layout of networks.** (A) PhyloSketch capture of a hybridization network representing lizard evolution, based on [Fig 2, 22], which was produced using ggnetworx [17]. (B) PhyloSketch layout of the network in a combining view. (C) PhyloSketch capture of a transfer network representing cat evolution, based on the image shown in [Fig S12E, 25], which was produced using PhyloNetworks [16]). (D) PhyloSketch layout of the network in a transfer view.

In this section, we focus on the *combining view*. A later section will describe how to adapt the algorithm to support the *transfer view*. We consider drawing rooted networks in a *left-to-right layout*—from the root on the left to the leaves on the right—and how to optimize such drawings.

We begin with the case of a *cladogram*, where edge lengths are ignored, and later extend the approach to *phylograms*, which incorporate edge lengths. Finally, we discuss how to adapt the algorithm to obtain a *circular layout* of the network. All algorithms described here use the following concept to calculate *y*-coordinates.

**Backbone tree.** For any reticulate node $v$ in $N$, we define the *lowest stable ancestor LSA(v)* to be the last node on all paths from the root $\rho$ to $v$. Consider the following operation on $N$: for each reticulate node $v$, set $u = LSA(v)$, delete all edges entering $v$, and create a new edge from $u$ to $v$. Note that application of this operation to all reticulate nodes gives rise to a rooted backbone tree $B$ whose root is $\rho$ and whose leaf set contains the leaf set of $N$, possibly along with other unlabeled leaves. Also note that the tree may contain through nodes. We call $B$ the *backbone tree* of $N$.

**Combining cladogram algorithm.** To compute a left-to-right layout of a rooted phylogenetic network $N$, we proceed in two main steps.

First, we perform a post-order traversal of the backbone tree $B$. For each leaf $u$ encountered, we assign $y(u) := i$, where $u$ is the $i$-th leaf visited. For each internal node $v$, we assign $y(v)$ either as the average of the $y$-coordinates of its children or, optionally, as the average of all its descendant leaves.

Second, we recursively compute the depth $d(v)$ of each node $v$ in the network $N$, defined as the maximum number of edges in any directed path from the root $\rho$ to $v$. We then set $x(v) := d(v)$ for all nodes.

Once coordinates have been assigned, we construct the left-to-right rooted cladogram $K(N, B, \mathcal{O})$ by drawing each edge $e = (v, w)$ as a two-segment path: from $(x(v), y(v))$ to $(x(v), y(w))$, then to $(x(w), y(w))$. Tree edges are rendered as right-angled paths, while reticulate edges may be drawn using quadratic curves, for example.

This algorithm yields an *early layout*, in which branching nodes are positioned as close to the root as possible (see Fig 2B). To produce a *late layout*, where branching nodes appear nearer to the leaves (see Fig 2D), a post-processing step can be applied that pushes nodes toward the leaves, keeping source nodes before target nodes.

## 2.3 Layout optimization

We define the *reticulate displacement* of a left-to-right drawing as

$$RD(K(N, B, \mathcal{O})) = \sum_{e=(v,w)\in R} |y(v) - y(w)|,$$

where $R \subseteq E$ is the set of reticulate edges (but excluding transfer-acceptor edges, as introduced later). This depends solely on the $y$-coordinates assigned to nodes during the post-order traversal of the backbone tree $B$, which in turn depend on the order $\mathcal{O}$ in which the children of each node $v$ are visited.

Among all orderings $\mathcal{O}$ of $B$, we propose to use one that minimizes the reticulate displacement. We refer to this approach as the *displacement optimization (DO)* algorithm.

**NP-completeness.** The decision version of the reticulate displacement minimization problem is NP-complete. The input consists of a rooted tree and a threshold $k$, and the question is whether there exists an assignment $\mathcal{O}$ of child orders (i.e., a permutation of the children at each node) such that the resulting layout score is at most $k$. This problem lies in NP, since a candidate embedding can be described by a polynomial-size set of permutations and evaluated in linear time. Furthermore, the optimization version of the problem is NP-hard, implying that the decision version is NP-complete.

An instance of MinLA consists of an undirected graph $G = (V, E)$ with $n = |V|$ vertices, and the goal is to find a bijective assignment $f : V \rightarrow \{1, \ldots, n\}$ that minimizes the total edge cost

$$\text{cost}(f) = \sum_{(u,v) \in E} |f(u) - f(v)|.$$

To reduce this to our layout optimization problem, we extend $G$ to a rooted phylogenetic network $N$ as follows: We introduce a root node $\rho$ and create an edge from $\rho$ to every original node $v \in V$. The original edges are considered reticulate edges. The new edges makeup the backbone tree $B$.

In this construction, the total reticulate displacement of the reticulate edges corresponds exactly to the MinLA cost function applied to the layout determined by the order of children below $\rho$. Thus, any solution to our layout optimization problem for this network instance provides a solution to the MinLA instance, and vice versa. Since MinLA is NP-hard [19], it follows that minimizing total reticulate displacement in our problem is also NP-hard.

**Heuristic optimization.** To address this optimization in practice, we perform a pre-order traversal of $B$. For each node $v$ that is the lowest stable ancestor (LSA) of some reticulate node, that is, the last node that lies on all paths from the root to the reticulation, we seek a permutation of its children in $B$ that minimizes the total reticulate displacement. If the number of children is at most eight, we exhaustively consider all permutations. Otherwise, we perform a heuristic search by iteratively swapping pairs of children, using simulated annealing [20] to escape local minima (parameters: start temperature = 1000, end temperature = 0.01, 1000 iterations per temperature step, cooling rate = 0.95).

## 2.4 Transfer view

Let $N$ be a rooted phylogenetic network on $X$. In the absence of additional information, we assume by default that every reticulate node is a *combining node*, to be displayed in the combining view.

A reticulate node $v$ is called a *transfer node* if exactly one of its incoming edges is designated as the *transfer-acceptor edge*, and the remaining incoming edges are called *transfer edges*. (For example, in the extended Newick format, the transfer-acceptor edge is indicated using `##`, rather than `#`, for the corresponding incoming edge.)

**Transfer cladogram algorithm.** To account for transfer nodes and their edges, we modify the construction of the backbone tree $B$ as follows: Tree nodes and combining nodes are handled as in the standard approach. For each transfer node $v$, we retain only its designated transfer-acceptor edge and remove all other incoming edges. This adjustment ensures that any drawing of $B$ avoids edge crossings among tree edges and transfer-acceptor edges.

We then compute the $x$ and $y$ coordinates as described for the combing view in Sect 2.2, with one modification. To encourage the source and target nodes of a transfer edge $e = (v, w)$ to share the same $x$-coordinate, we process each such edge as follows: if $d(v) < d(w)$, we set $d(v) := d(w)$ and update the map $d$ accordingly for all affected nodes. This process is repeated multiple times to account for cases where transfer edges mutually influence each other. Finally, we set $y(v) := d(v)$ for all nodes $v$ in $N$.

## 2.5 Phylograms

So far, we have discussed drawing a cladogram, in which all leaves are placed on the right side of the drawing, all with the same horizontal coordinate 0, and any edge lengths, if given, are ignored.

To compute a *phylogram* of $N$, we assume that every tree edge or transfer-acceptor edge $e \in E$ is assigned a non-negative weight $\omega(e)$. The algorithm described here can be applied to both the combined view and the transfer view [26].

**Combine and transfer phylogram algorithm.** We first use the approach described in Sect 2.2 to compute all $y$-coordinates of the nodes, followed by the layout optimization strategy from Sect 2.3.

To compute the $x$-coordinates, we assign the root $\rho$ the coordinate $x(\rho) = 0$ and then perform a breadth-first traversal as follows:

Let $v$ be a node for which all parent nodes in $N$ have already been processed. There are two cases to consider:

- Tree or transfer node: Then $v$ has an incoming edge $e = (u, v)$ that is either a tree edge or a transfer-acceptor edge and we set $x(v) = x(u) + \omega(e)$.
- Combining reticulation node: Then $v$ is a combining node with $k \geq 2$ incoming reticulate edges $e_1 = (u_1, v), \ldots, e_k = (u_k, v)$, and we set $x(v) = \max\{x(u_1), \ldots, x(u_k)\} + \delta$, where $\delta$ is a small, fixed value that is used to ensure that reticulate edge always go from left-to-right.

### 2.6 Circular layout

A circular cladogram or phylogram can be derived from a left-to-right layout by converting $y$-coordinates, which range from 0 to $n-1$, into angles ranging from 0° to $\frac{n-1}{360}°$, and by computing radii instead of $x$-coordinates, see Fig 3.

Optimization of the circular drawing can be achieved by optimizing the underlying left-to-right layout, as described above, using a slightly modified cost function: For each reticulate edge $e = (v, w)$, the cost is defined as

$$\min\{|y(v) - y(w)|,\ H - |y(v) - y(w)|\},$$

where $H$ is the number of leaves in $B$, plus 1. This accounts for the fact that in a circular layout, the shortest path between two nodes may go around the circle in either direction.

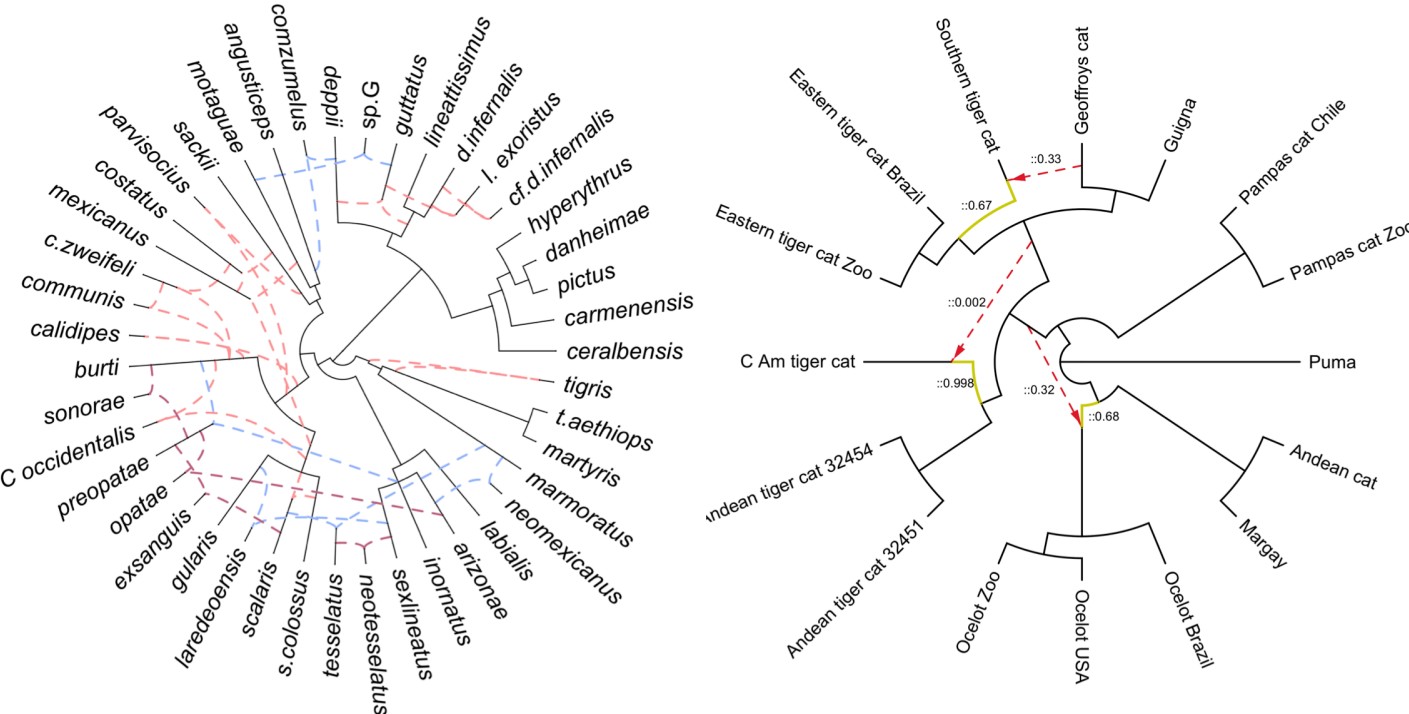

**Fig 3. Circular layout.** (A) PhyloSketch circular combining view, network from [Fig 2, 22]. (B) PhyloSektch circular transfer view, network from [Fig S12E, 25].

## 2.7 Equal spacing of leaves

The cladogram and phylogram layout algorithms presented above suffer from the deficiency that leaves may not appear equally spaced, due to the presence of unlabeled leaves in the backbone tree $B$ of the network $N$.

To address this issue, once an optimal layout has been determined, we perform the following post-processing step: We repeat the post-order traversal of the backbone tree $B$ to recompute $y$-coordinates. This time, we assign integer values $0, 1, \ldots$ to the labeled leaves and assign intermediate fractional, equally spaced coordinates to the unlabeled leaves. Some existing approaches fail to address this problem, as is evident in Fig 2C and in Fig 4C.

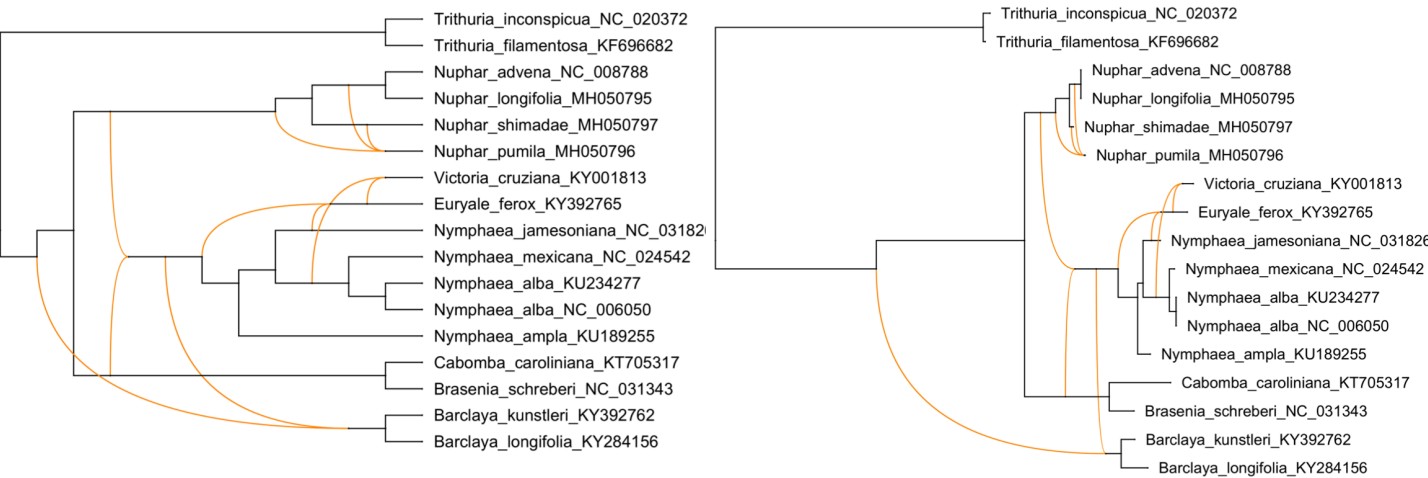

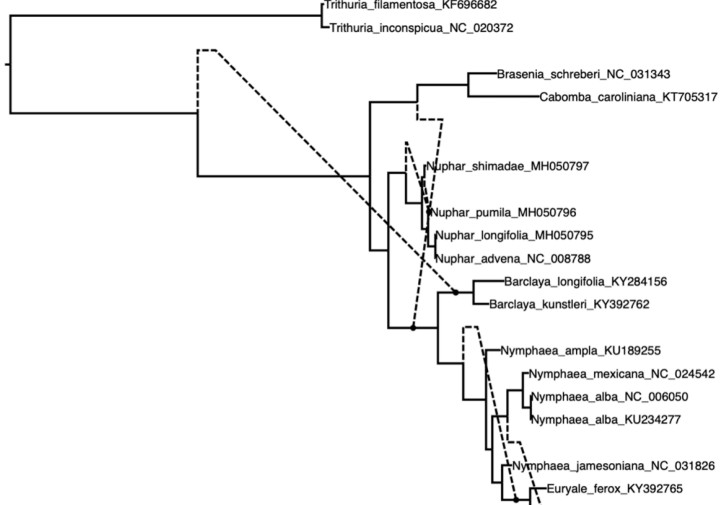

**Fig 4**. **Cladogram, phylogram and IcyTree visualization.** A rooted phylogenetic network obtained by applying PhyloFusion [9] to 11 NADH dehydrogenase-associated gene trees of water lilies [27], is shown here as (A) a combining cladogram and (B) a combining phylogram, computed using the described algorithms. Note that the vertical spacing of leaves is uniform. In contrast, (C) an "ancestral recombination graph" visualization computed by IcyTree [15] exhibits gaps in the vertical spacing.

## 3 Implementation

We provide implementations of all introduced algorithms, including the transfer and combine views, left-to-right and circular layouts of cladograms and phylograms, and reticulate-displacement based layout optimization.

### 3.1 PhyloSketch App

The *PhyloSketch* App is an interactive tool for working with phylogenetic trees and rooted networks. Trees and networks can be imported and exported using the extended Newick format. By default, reticulate nodes are interpreted as combining nodes; however, users can interactively declare transfer-acceptor edges, thereby reinterpreting the corresponding reticulate nodes as transfer nodes.

*PhyloSketch* also supports the interactive construction and editing of trees and networks. When a user draws a path using a mouse or touch screen, the software attempts to interpret the path as an edge, connecting its start point to a nearby node or an interior point of an existing edge, or its end point to another nearby node or interior point, if possible. A proposed path is accepted as a new edge only if it will not introduce a directed cycle into the network.

The program additionally provides a range of operations for modifying the topology of the tree or network, as well as tools for styling and labeling nodes and edges.

In Fig 5, we show that layout computations in PhyloSketch complete in acceptable time, although they are slower than in Dendroscope. However, the new algorithm produces layouts with substantially lower displacement values than the algorithm implemented in Dendroscope.

### 3.2 Network capture

The *PhyloSketch* App includes a workflow that assists users in capturing a rooted phylogenetic tree or network directly from an image (see Fig 2A and 2C). To capture a phylogenetic tree or network, the user loads an image into the program and then specifies the location of the root. The workflow then proceeds as follows:

1. The Tesseract library [28] is used to detect labels in the image.
2. A skeletonization algorithm [29] reduces all lines to single-pixel width.
3. All nodes (i.e., branching points and endpoints of lines) are detected using a pixel mask.
4. A flood-fill algorithm is applied to identify all paths connecting pairs of nodes.
5. Paths that lie entirely within the bounding box of a detected label are removed.
6. A root node is created at the chosen root location, splitting any nearby path if necessary.
7. The layout orientation, such as left-to-right, top-to-bottom, or radial, is determined by comparing the indicated root location with the positions of detected nodes.
8. Detected labels are associated with nearby nodes, taking the root location into account.
9. In order of increasing distance from nodes added so far, each remaining path is considered a potential new edge, and an attempt is made to incorporate it, as described in the previous section.

This approach assumes that the tree or network is drawn in dark colors on a white background, that edges are rendered as reasonably thin lines, and that nodes are small. Also, the presence of labels overlapping the the nodes or edges of the phylogeny, or additional lines indicating annotations, etc, will have a negative impact on performance. In practice, complex or poor-quality images will require preprocessing in third-party image software.

The capture of phylogeny from images will usually require editing by the user especially when reticulation edges cross other edges. Several post-processing steps are available to clean-up the captured phylogeny: "through nodes" (of in-degree and out-indegree 1) can be removed; nodes of in-degree two and out-degree two can be replaced by crossing

A

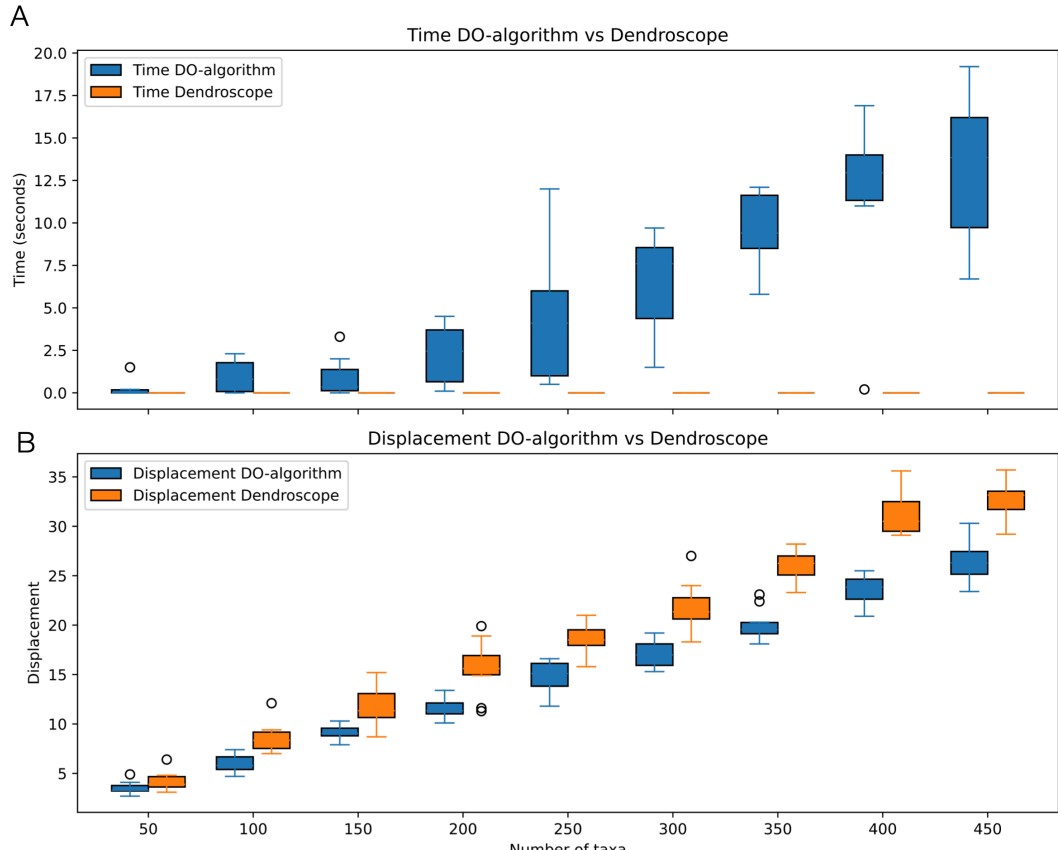

B

Fig 5. **Comparison with Dendroscope.** Using randomly generated rooted phylogenetic networks with $n = 50, \ldots, 450$ taxa and $h = 0.2 \times n$ reticulations (10 replicates each), we compared the performance of the new displacement optimization (DO) algorithm with that implemented in Dendroscope. (A) Wall-clock time (in seconds) on a MacBook Pro (M4 processor) to compute and optimize a combined rectangular cladogram. (B) Total reticulate displacement for both methods, normalized by the total height of the drawing.

edges; edge directions can be reversed; and sets of nodes can be merged into a single node while maintaining labels and connectivity.

## 4 Discussion

While many programs and packages exist for computing, constructing, and editing phylogenetic trees, and to a lesser extent, phylogenetic networks, to the best of our knowledge, *PhyloSketch* is the first program that allows users to interactively create a phylogenetic tree or network from scratch (see Fig 6), or to capture (parts of) a phylogenetic network from an image with the help of image-processing algorithms.

The proposed algorithm for computing a left-to-right cladogram builds on the approach described in [26] and implemented in *Dendroscope* [14], but differs in how the layout cost is computed and in its use of simulated annealing for optimization, resulting in layouts with lower displacement. Unlike the Dendroscope algorithm, our method honors transfer–acceptor edges and prevents crossings between these and tree edges. In addition, it explicitly seeks to place the source and target of transfer edges on the same level whenever possible. Furthermore, we distinguish between and provide both early and late cladograms.

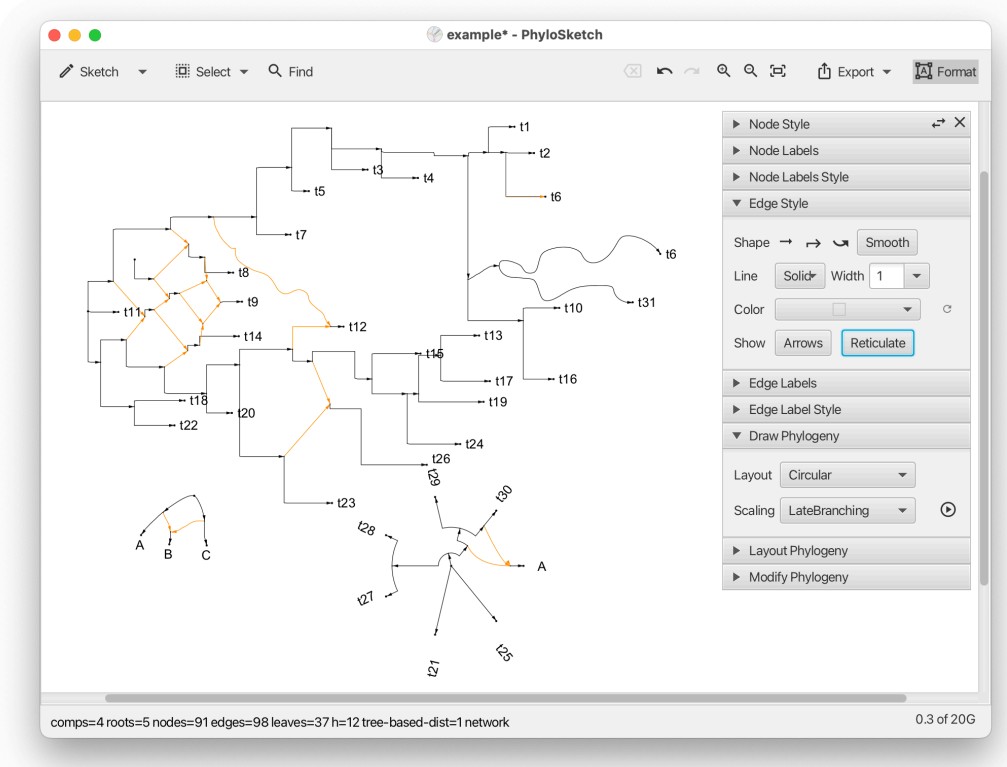

**Fig 6**. **PhyloSketch.** This shows the main window of the app, with several nodes and edges that have been interactively sketched and labeled. A late-branching circular layout has been applied to the bottom nework.

Using the backbone tree to compute and optimize network layouts provides a useful simplification. However, this approach can miss "obvious" improvements that would require rearrangements not represented in the backbone tree. For example, in Fig 4A, switching the subtree containing the two lowest taxa (*B. kunstleri* and *B. longifolia*) with the one above it (containing *C. caroliniana* and *B. schreberi*) would significantly reduce the reticulate displacement of the two associated reticulation edges. However, this rearrangement is never considered since the subtrees are not attached to the same node in the backbone tree.

Other tools that provide visualizations of rooted reticulate networks include Dendroscope [14], PhyloNetworks [16], IcyTree [15] and ggnetworx [17]. While the first of these provides both combining and transferring views, the latter two both provide only a transfer view and both suffer from a non-uniform spacing of leaves (see Fig 2C and Fig 4C, respectively). The network layout algorithms described here are now also used in SplitsTree [18].

Much effort was invested in designing a simple yet powerful user interface for *PhyloSketch*. Future work may include extending the layout and capture algorithms to support other classes of networks, and providing this software as an iOS app.

## Acknowledgments

The author is thankful to Elizabeth Allman and Mike Steel for many helpful discussions.

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
