## [Decision Letter · Decision Letter 0]

20 Oct 2025

PCOMPBIOL-D-25-01507

Sketch, capture and layout phylogenies

PLOS Computational Biology

Dear Dr. Huson,

Thank you for submitting your manuscript to PLOS Computational Biology. After careful consideration, we feel that it has merit but does not fully meet PLOS Computational Biology's publication criteria as it currently stands. Therefore, we invite you to submit a revised version of the manuscript that addresses the points raised during the review process.

Please submit your revised manuscript within 60 days Dec 20 2025 11:59PM. If you will need more time than this to complete your revisions, please reply to this message or contact the journal office at ploscompbiol@plos.org. Please include the following items when submitting your revised manuscript:

We look forward to receiving your revised manuscript.

Kind regards,

Sebastian Duchene

Academic Editor

PLOS Computational Biology

Feilim Mac Gabhann

Editor-in-Chief

PLOS Computational Biology

**Additional Editor Comments (if provided):**

Dear Prof Huson,

Thanks for submitting your article to Plos Computational Biology. It has been reviewed by two experts, both of which are very positive about this software and algorithm. They have made a few suggestions with which I agree and I therefore invite you to address them and resubmit your manuscript. I am recommending 'major revisions', but I do not anticipate that these comments warrant substantial additional analyses.

In particular, I deem that it is important to play close attention to the reviewer's comments about the parameters users need to fix to optimise algorithm, and some improved functionality (or some sort of warning) for larger trees.

Best wishes,

Sebastian Duchene

**Journal Requirements:**

**Reviewers' comments:**

Reviewer's Responses to Questions

**Comments to the Authors:**

Reviewer #1: Please see the attached file.

Reviewer #2: Uploaded as attachment

**Have the authors made all data and (if applicable) computational code underlying the findings in their manuscript fully available?**

Reviewer #1: Yes

Reviewer #2: Yes

PLOS authors have the option to publish the peer review history of their article (what does this mean?). If published, this will include your full peer review and any attached files.

Reviewer #1: **Yes: **Wytamma Wirth

Reviewer #2: **Yes: **Camila Duitama González

**Figure resubmission:**
---

## [Editor Report · Decision Letter 1]

1 Dec 2025

Dear Prof. Huson,

We are pleased to inform you that your manuscript 'Sketch, capture and layout phylogenies' has been provisionally accepted for publication in PLOS Computational Biology.

Best regards,

Sebastian Duchene

Academic Editor

PLOS Computational Biology

Feilim Mac Gabhann

Editor-in-Chief

PLOS Computational Biology

Dear Prof Huson,

Thanks for revising your manuscript according to comments raised by two reviewers. You have addressed all of their comments and I at this stage I am delighted to accept your paper for publication in Plos Computational Biology. PhyloSketch is a very useful tool that adds novel capabilities to the existing toolkit (particularly editing and visualising networks in a publication-ready format).

I thank you again for submitting to our journal and I look forward to seeing your manuscript published.

Best wishes,

Sebastian

---

## [Editor Report · Acceptance letter]

PCOMPBIOL-D-25-01507R1

Sketch, capture and layout phylogenies

Dear Dr Huson,

I am pleased to inform you that your manuscript has been formally accepted for publication in PLOS Computational Biology. Your manuscript is now with our production department and you will be notified of the publication date in due course.

With kind regards,

Judit Kozma
